# Hepatitis B Virus DNA Integration: In Vitro Models for Investigating Viral Pathogenesis and Persistence

**DOI:** 10.3390/v13020180

**Published:** 2021-01-26

**Authors:** Thomas Tu, Henrik Zhang, Stephan Urban

**Affiliations:** 1Storr Liver Centre, Faculty of Medicine and Health, Westmead Clinical School and Westmead Institute for Medical Research, The University of Sydney, Westmead, NSW 2145, Australia; hzha8212@uni.sydney.edu.au; 2Centre for Infectious Diseases and Microbiology, Marie Bashir Institute for Infectious Diseases and Biosecurity, University of Sydney at Westmead Hospital, Westmead, NSW 2145, Australia; 3Department of Infectious Diseases, Molecular Virology, Heidelberg University Hospital, Im Neuenheimer Feld 345, 69120 Heidelberg, Germany; Stephan.Urban@med.uni-heidelberg.de; 4German Center for Infection Research (DZIF), Heidelberg Partner Site, Im Neuenheimer Feld 345, 69120 Heidelberg, Germany

**Keywords:** hepatitis B virus, HBV DNA integration, hepatocellular carcinoma (HCC), non-homologous end joining, microhomology-mediated end joining, HBV double-stranded linear DNA, viral persistence

## Abstract

Hepatitis B virus (HBV) is a globally-distributed pathogen and is a major cause of liver disease. HBV (or closely-related animal hepadnaviruses) can integrate into the host genome, but (unlike retroviruses) this integrated form is replication-defective. The specific role(s) of the integrated HBV DNA has been a long-standing topic of debate. Novel in vitro models of HBV infection combined with sensitive molecular assays now enable researchers to investigate this under-characterised phenomenon with greater ease and precision. This review covers the contributions these systems have made to understanding how HBV DNA integration induces liver cancer and facilitates viral persistence. We summarise the current findings into a working model of chronic HBV infection and discuss the clinical implications of this hypothetical framework on the upcoming therapeutic strategies used to curb HBV-associated pathogenesis.

## 1. Introduction

Chronic infection with the human hepatitis B virus (HBV) is one of the major drivers of liver disease and is the most common cause of liver cancer worldwide. Approximately one third of the world’s population has been exposed to the virus, and ~257 million people currently live with a chronic HBV infection [1,2]. Chronic HBV infections are usually life-long (with few exceptions) as the virus has developed several persistence mechanisms to escape immune surveillance, including replication via a highly-stable nuclear episomal template (cccDNA) mimicking a mini-chromosome. Another mode of escaping elimination by the immune systems is the expression of sub-viral particles that dampen antiviral immune responses by mechanisms not yet fully understood.

### 1.1. Natural History of Chronic Hepatitis B 

Chronic HBV infections are generally asymptomatic for long periods (up to decades) as the virus replicates without triggering an antiviral immune response [3]. Decades later, a sub-optimal immune response is raised against HBV-expressing hepatocytes through yet-unknown mechanisms. The host response is usually inadequate to overcome viral persistence mechanisms, but instead leads to chronic inflammation, liver damage, and hepatic disease. 

Surveillance and ongoing care for chronic HBV infection is complicated by psychosocial factors, including limited access to or distrust of the health care system, financial burdens, social stigma, and systemic discrimination [4]. A poor cascade of care results: only ~10% of cases are diagnosed and only ~3% are adequately treated even in some developed countries [5,6]. Thus, HBV-related disease generally progresses unmonitored and kills ~880,000 people annually through liver cirrhosis and liver cancer [1,2]. 

The mechanisms by which HBV causes liver cancer are still not well understood and are under intense investigation. One of the major factors thought to be involved in hepatocarcinogenesis is the integration of the viral DNA genome into cellular chromosomes. This reportedly induces pro-oncogenic pathways through several means (previously reviewed in detail [7,8,9]), including *cis*-mediated mechanisms (e.g., HBV DNA integrations modulating expression of proximal cellular genes) or *trans*-mediated mechanisms (e.g., chronic expression of particular viral antigens). This review focuses on how in vitro models have helped in furthering our understanding of how integrated HBV DNA contributes to virus persistence and pathogenesis.

### 1.2. HBV Structure 

HBV is the prototypical member of the Hepadnaviridae family, which are small enveloped, hepatotropic DNA viruses (~3.2 kbp) replicating via reverse transcription. The double-stranded DNA genome of HBV can take two forms (Figure 1): relaxed circular DNA (rcDNA, generally present in ~90% of virions [10]) or double stranded linear DNA (dslDNA, ~10% of virions). Unlike the fully replication-competent relaxed circular form, the dslDNA-containing viral particles are replication-defective (detailed further below) but can integrate via non-homologous recombination into hepatocyte chromosomes. The HBV genome contains four overlapping open reading frames that encode for seven viral proteins, including: the HBV core antigen (HBcAg, which forms the viral capsid), e antigen (HBeAg, a secreted viral protein encoded by the core/pre-core ORF with reported tolerogenic properties [11]), surface protein (HBsAg, which has three forms large, medium and small—or LHBs, MHBs, and SHBs—that envelop the virion), polymerase (HBV pol, which is essential for viral reverse transcription and replication), and the x protein (HBx, which regulates transcription of viral genes from the cccDNA episome). 

### 1.3. HBV Replication 

HBV replication starts with rcDNA-containing HBV particles attaching to heparan sulphate proteoglycans on the surface of hepatocytes [12,13,14] (Figure 2). This attachment sets up conditions for a high-affinity interaction between the preS-domain of LHBs and sodium taurocholate co-transporting polypeptide (NTCP), which acts as the functional receptor for HBV [15,16]. Binding is reportedly followed by clathrin-mediated endocytosis of the viral particle [17,18] and cytoplasmic release of the HBV nucleocapsid, which is subsequently transported to the nucleus and disassembles at the nuclear pore complex [19]. 

The HBV rcDNA genome is released into the nucleus, where it is repaired into the covalently closed circular DNA (cccDNA) form by multiple host DNA repair enzymes [24,25,26,27,28,29,30,31]. cccDNA acts as the template for all viral transcripts including pregenomic RNA (pgRNA), which is reverse-transcribed in nucleocapsids formed by the viral core antigen. This results in HBV DNA-containing nucleocapsids that are enveloped by host membranes studded with all three forms of HBsAg and then secreted from the cell via multi-vesicular bodies as virions. 

### 1.4. HBV dslDNA and HBV DNA Integration

While the majority of intra-capsid reverse transcription events result in the formation of rcDNA, a minority of virions contain HBV dslDNA (though this ratio can fluctuate in patients in different stages of infection [10]). The production of dslDNA is dependent on binding and circularisation signals present on the reverse-transcribed strand of HBV DNA [32,33]. When dslDNA-containing virions infect a cell, HBV genome (instead of being converted into cccDNA) can integrate into cellular genome at the site of cellular double-stranded DNA breaks [34]. 

The role of integrated HBV DNA is currently unclear: unlike retroviruses, the integrated form of HBV is replication-deficient because of rearrangements that abrogate the expression of capsid, polymerase and functional pgRNA (Figure 1). Firstly, the orientation of the HBV dslDNA separates the HBcAg and pol at its 5′ end from their native promoter at its 3′ end. Moreover, the poly A signal shared by all viral transcripts is located on the 5′ end instead of the 3′ end of the dslDNA, leading to truncated transcripts lacking a canonical poly A. Most viral transcripts expressed from integrated HBV DNA can retain function by terminating using a non-canonical poly A signal retained at its 3′ end [35,36]. However, pgRNA is rendered non-functional as the premature termination removes structural elements from the 3′ end necessary for initiating pgRNA reverse transcription (namely, the direct repeat 1 and phi sequences [37,38,39,40]). Finally, terminal truncations on both ends of the integrated genome are introduced by the error-prone host DNA repair pathways during HBV DNA integration and can disrupt viral ORFs [34,41,42,43]. Together, these effects render integrated HBV DNA replication-deficient. No known cell lines with integrated HBV DNA derived from a natural infection produce infectious virus, though they can still express functional HBsAg [44,45,46,47].

Various functions have been ascribed to the integrated HBV DNA form (e.g., supporting viral persistence and pathogenesis). In vitro models (Table 1) have enabled detailed understanding of these aspects, including: the molecular mechanisms governing HBV DNA integration; its role as a source of HBsAg in a chronic infection; and how it is involved in HCC initiation and progression. The remainder of this review summarises the knowledge that these model systems have provided the field.

### 1.5. In Vitro Models of HBV DNA Integration

In vitro models vary in their ease of use (or reproducibility) in integration studies and inversely how closely they resemble integrations from a true infection in people with HBV. Tumour-derived cell lines from HBV patients contain integrations formed in the native setting but cannot be used to understand underlying mechanisms as they do not generate new integrations. Conversely, nuclear introduction of large numbers of HBV DNA molecules (via over-expressing constructs or transfection) can drive relatively high integration rates, allowing easy study of the integration process. However, this does not reflect a true infection; nuclear import of new HBV virions occurs relatively inefficiently in HBV infection [48,49,50,51], so levels of HBV DNA in infected hepatocytes is relatively low. Newly-developed infection models expressing the HBV receptor have solved this to some degree by largely recapitulating the virological aspects accurately. However, the low integration rate is challenging to detect without specialised methods, which can be difficult to reproduce. In essence, all studies need to be interpreted with these limitations in mind.

**Table 1 viruses-13-00180-t001:** A summary of in vitro model systems for human HBV DNA integration.

Model System	Type	Infectious HBV Produced?	Forms New HBV DNA Integrations?	Refs
PLC/PRF/5	Tumour-derived cell line ^1^	No	No	[44,45]
Hep3B	Tumour-derived cell line ^1^	No	No	[46,47]
HepG2.2.15	Engineered HBV-producer cell line ^2^	Yes	Yes	[52,53]
HepAD38	Engineered HBV-producer cell line ^2^	Yes	Unknown	[54]
Transfection of HBV over-length constructs	HBV transfection ^3^	Yes	Unknown	[55,56,57,58,59]
Transfection of HBV monomeric DNA	HBV transfection ^3^	Yes	Unknown	[3,60,61]
Transfection of HBV virion DNA	HBV transfection ^3^	Yes	Yes	[62]
Transfection of in vitro transcribed HBV pgRNA	HBV transfection ^3^	Yes	Unknown	[63]
Huh7-NTCP	HBV infection ^4^	Yes	Yes	[48,51]
HepG2-NTCP	HBV infection ^4^	Yes	Yes	[48,51]
HepaRG	HBV infection ^4^	Yes	Yes	[48,51]
HepaRG-NTCP	HBV infection ^4^	Yes	Yes	[48,51]
Primary human hepatocytes	HBV infection ^4^	Yes	Yes	[48,51]

^1^ Tumour-derived cell lines contain replication-deficient integrated HBV DNA acquired during infection. ^2^ Engineered HBV-producer cell lines contain stable recombinant replication-competent HBV constructs. ^3^ HBV transfection models involve ectopic introduction of viral DNA or RNA constructs. ^4^ HBV infection models involve *bona fide* viral infection of cells expressing the viral receptor NTCP.

## 2. Molecular Mechanisms of HBV DNA Integration 

### 2.1. HBV DNA Integration Occurs upon de Novo Infection

Integrated HBV DNA was initially described in liver cancers (or cell lines derived from these) and regenerative hepatic nodules in cirrhotic liver tissue [43,64,65,66,67,68,69,70,71,72,73,74,75]. Given these patients were in late disease progression, these observations were initially interpreted as HBV DNA integration being a phenomenon associated with cellular genotoxic damage and chromosomal instability. However, the relatively insensitive method used (Southern blot hybridisation) complicates this interpretation: clonal expansion (driven by chronic inflammation and hepatocyte transformation) results in multiple copies of the same integration which facilitate their detection, but integrations occurring in earlier infection require more sensitive methods. 

A fuller picture of HBV integration was revealed when tissues were analysed earlier in infection and with specifically designed PCR-based assays with higher sensitivity (methods for detecting HBV DNA integrations have been reviewed previously [76]). HBV integration was shown to occur in patients in all phases of chronic infection [77,78], in acute infection [79] and in children [80,81]. Moreover, in vivo (woodchuck and duck models of HBV infection [34,82]) and in vitro HBV models showed that integration occurs within days [48,51] (on the order of tens of minutes by some reports [83]). 

While stable cell lines that overexpress HBV can force new DNA integrations [52,53], integration does not seem to occur continuously in a persistently-infected cells. In vitro HBV infection models have shown that integration is not affected by inhibition of reverse transcription (e.g., from nucleoside analogue treatment [48]) and requires neither *de novo* HBV replication [51] nor viral proteins of the incoming virus [62]. Together, this suggests that integration occurs at the initial infection and that additional integration events as results are due to new infection events or clonal expansion.

### 2.2. Molecular Pathways Involved in Integration

In vitro models have shown that it is the dslDNA form of HBV that integrates into the host genome. In a foundational study of an in vitro duck HBV model, Bill and Summers introduced a transgene coding for the I-*Sce*I restriction enzyme into a chicken hepatoma cell line containing a single I-*Sce*I recognition site [41]. They found integration of viral DNA at the restriction site after transfection with a duck HBV over-length expression construct. Integration rates increased after cells were transfected with a construct encoding a viral mutant that generated more dslDNA. 

This study was also pivotal in showing that hepadnaviral DNA integration occurs at double-stranded DNA breaks in the cellular genome. These results are consistent in human HBV models: increased integrations were observed in HepG2.2.15 cells after treatment with Poly(ADP-Ribosyl)ation inhibitors (which slow down the repair of DNA breaks) or hydrogen peroxide (which increase oxidative DNA damage) [52,53]. 

Analysis of the termini of the integrated virus genomes have found frequent short deletions, suggesting the involvement the error-prone non-homologous end joining (NHEJ) DNA repair pathways. The majority of virus-host junctions observed in patient tissues and in vitro infection models show consistent results [41,48]. Moreover, little to no homology between host and viral DNA is observed, indicating NHEJ repair pathways. Interestingly, a subset of HBV DNA integrations shows short runs of homology between the virus and host sequences (2–10 bp), suggesting some events may occur through the alternate NHEJ pathway (or micro-homology mediated end-joining) [23,48,84,85,86]. 

Supporting this case in a true infection model, integration can also occur in terminally-differentiated HepaRG-based cells as well as non-dividing primary human hepatocytes [48]. As NHEJ repair pathways are preferred in G0/G1 phase (due to the poor accessibility of sister chromatids to participate in homology-mediated DNA repair), this strongly suggests NHEJ involvement in the HBV DNA integration process. This is consistent with data from in vitro transfection models showing that integration (unlike with retroviruses) occurs without the input of viral protein-mediated pre-integration complexes [62], suggesting that integration is entirely dependent on these cellular repair pathways. 

### 2.3. Sites of Integration in the Host Genome

Another strong indicator of NHEJ pathway involvement is the largely random distribution of integrations throughout the host genome. Unlike many retroviruses that preferentially integrate into transcriptionally active regions in the host genome, HBV does not appear to have any major favoured sites of integration with respect to any cellular structures or functional motifs (in both in vitro and non-tumour patient tissues, as previously reviewed [76]). Our studies have been generally consistent with this but observed slight enrichment (~2-fold greater frequency above a random in silico simulation of integration) in broad vicinity of cellular CpG islands, coding regions, and promoter sites [23]. Given this enrichment was observed in both patient tissues and directly after in vitro infection, this suggested that liver disease progression and associated clonal expansion in the patients did not select for integrations in these regions. Instead, our use of in vitro models suggested that the underlying molecular mechanisms targeted integrations to these sites (possibly due to more DNA breaks occurring in chromatin with fewer histones bound). 

By contrast, the HBV DNA integration sites in HCC tumour tissue are enriched in particular genetic regions (particularly cancer associated genes), though these likely arise due to selection towards integration events that induce cancer progression (as explained in the next section).

## 3. The Role of Integration in HBV-Associated HCC

There is a clear association between integrated HBV DNA with liver cancer; the frequency of HBV integrations in HBV-associated cancers (1–5 integrations in 60–90% of all tumours [87]) is much higher than expected if a hepatocyte was picked at random to initiate the tumour from the non-tumour liver (where the integration frequency is ~1 per 1000–10,000 cells [48,78]). Despite this strong association, it has been difficult to untangle the true relationship between integration and cancer. 

### 3.1. Cis-Mediated Mechanisms

Integrated HBV DNA forms have viral promoters on their 3′ ends (the HBc and HBx promoters) that can drive downstream cellular transcription, which can in turn drive carcinogenesis. Indeed, this appears to be the case in some HBV-associated HCCs, in which integrations repeatedly occur close to the promoter of the human telomerase reverse transcriptase (hTERT), or within mixed-lineage leukaemia 4 (MLL4) or cyclin e1 (CCNE1) genes [84,88,89,90,91]. 

However, these represent a minority of HCCs, with most having integrations in sites not associated with cancer. Moreover, insertions at these sites are not likely to be sufficient to initiate cancer. In vitro cell models have shown that while over-expression of hTERT can immortalise some human hepatocytes, these cells are unable to induce cancer formation even when injected into immune-deficient mice [92,93,94]. Thus, alternate mechanisms are needed to explain how HBV infection initiates HCC. 

### 3.2. Trans-Mediated Mechanisms

Another reported mechanism for HBV-associated HCC development is transformation by viral antigens. However, HBV proteins in themselves do not appear to be carcinogenic; transgenic mice expressing HBV antigens from native promoters do not show any liver pathohistology [95,96]. Moreover, patients in immune tolerant phase (who express very high levels of virus protein) are at lower relative risk of developing HCC, particularly compared to patients in later phases in whom viral expression is much lower [97]. Finally, HBV cccDNA is efficiently depleted by the mitosis of infected cells [50], which would limit cancer progression after initial expansion of a transformed sub-clone. 

A possible mechanism could be the altered expression of viral proteins (either wild-type or mutated) from the integrated HBV genomes. Expression of wild-type HBx protein alone leads to an increased risk of HCC development in older mice [98], but does not spontaneously induce HCC without an additional source of liver damage. Many other reports of HBx-induced HCC in mouse models have been driven by highly-active promoters (likely not at physiological levels of HBx) and so must be interpreted with caution [99]. Similarly, over-expression of HBsAg from a highly transcriptionally-active albumin promoter spontaneously induces pre-neoplastic lesions and HCC in transgenic mice [100], while mice expressing HBsAg alone from native promoters maintain histologically normal livers [101,102].

The integrated HBV DNA leads to C-terminal truncation of the HBx ORF, which may be involved in carcinogenesis. A recent report has described the C-truncated HBx variants in HCC and found that they retain the ability to interact with the E3 Cul4A/DDB1 complex (previously shown to be important in modulating of transcription of HBV and likely host genes [103]), while losing the domain responsible for its anti-proliferation effects [104]. This disruption to the internal balance of HBx’s functions could drive HCC in cells that chronically express it from integrated HBV DNA. 

Integrated HBV DNA genomes can also contain mutations in the HBsAg region. The PLC/PRF/5 hepatoma-derived cell line contain some of the few fully-sequenced integrated HBV DNA genomes; some of these show mutations, though it is unclear how representative these are for integrated forms in non-tumour tissue. One of these integrations includes a deletion in the HBsAg ORF [105]. Mutations in HBsAg (particularly in the PreS region) have been linked to endoplasmic reticulum stress (both in vitro [106,107] and in animal models [108,109]) and there is evidence that these cells clonally expand in patient livers, suggesting it confers a survival advantage to the cells [106,110,111]. These HBsAg mutants and the “ground-glass” histological change they induce in liver tissues are indeed more frequent in patients with HCC [112,113,114], though exact mechanisms behind this association remain to be fully characterised. 

## 4. Integrated HBV DNA as a Source of HBV Surface Antigen 

Clearance of HBs is considered an end-point for HBV therapy (known as functional cure). In this state, HCC risk is significantly reduced, and liver fibrosis is halted and can even reverse. Moreover, high levels of circulating HBsAg have been linked to virus-specific tolerance [115,116] and the reduction of HBsAg can restore antiviral T-cell responses to clear HBV-infected hepatocytes (shown previously by siRNA knock-down of HBsAg in a HBV-expressing mouse model [117]).

Notably, integrated HBV DNA can be a source of HBsAg. While the episomal replication-competent cccDNA form of HBV can code for HBsAg, early studies of the hepatoma-derived cell lines Hep3Bs [46,47] and PLC/PRF/5 [44,45] showed that integrated HBV DNA can also act as a template for HBsAg expression. Indeed, the HBsAg expressed by PLC/PRF/5 can package and secrete infectious HDV particles [118], showing that these proteins can be fully functionally active. Indeed, recent studies have shown integrated HBV DNA as the major source of HBsAg in HBeAg-negative patients (i.e., in later phases) by RNAseq analysis of human liver tissues [36,119].

### 4.1. A Hypothetical Model of HBV DNA Integration Dynamics during Chronic Infection

Using the discoveries described in this review, we have developed a hypothetical model of HBV integration dynamics and how it can act as a stable source of HBsAg throughout chronic infection (Figure 3). While many models have been developed to try to understand viral-host dynamics [120], none of these appear to have incorporated the effects of HBV DNA integration, clonal expansion and other new phenomena that have been discovered in recent studies.
The virus infects the whole liver following the initial infection. In this initial phase, the majority of HBsAg is derived from cccDNA, as integrated HBV DNA is rare (1 integration per ~10,000 cells, as shown in in vitro [48,51] and in vivo models). Ongoing nuclear import and integration in chronically-infected cells is rare [48,49,50,51], so integration rate remains relatively stable. New infections (which contribute to new HBV integrations) are suppressed by super-infection exclusion effected through LHBs expression by infected cells (discovered in in vitro models of duck HBV infection [121]).Upon activation of the anti-HBV immune response, HBV e antigen and HBV polymerase (which are coded by cccDNA, but not integrated HBV DNA) are the main antigens that are targeted [122].cccDNA-containing hepatocytes are targeted for cell death and replaced by mitosis of neighbouring hepatocytes. This facilitates the clonal expansion of hepatocytes containing integrations (experimentally observed in human liver tissues [77,78]).In parallel, replicative space increases due to loss of cells expressing HBsAg from cccDNA. cccDNA is lost with mitosis [48,50,123] and daughter cells are susceptible to reinfection (and therefore new integration events). Cells infected by rcDNA-containing virions that form cccDNA are selected against, while cells infected by dslDNA-containing viruses and integrate are not subject to this selection pressure.The ratio of cccDNA to integrated HBV DNA frequency therefore decreases. Correspondingly, the ratio of both HBV e antigen and serum HBV DNA to HBsAg decreases (as observed in patient sera [124]).A new equilibrium emerges where the level of immune suppression by HBsAg [115] counteracts the immune stimulation by productively-infected hepatocytes, reducing the amount of HBeAg secreted and intrahepatic cccDNA.Long-term exposure to the stably high levels of HBsAg can deplete HBV-targeting T-cells [116]. In this way, HBV integration maintains a persistent infection.


There are several aspects to this theoretical model that require further experimental support. Mathematical modelling would be ideal in determining if emergent behaviour from these initial assumptions and multiple interacting factors reflects clinical observations of HBV patients. To provide realistic ranges for parameters, precise experiments must be carried out to define the effect sizes of each of these factors (e.g., immunosuppression by HBsAg, immune-stimulation by HBeAg-expressing cells). This may require a firmer understanding of anti-HBV immunity than the field currently has. 

Even relatively simple parameters, such as cccDNA per hepatocyte, remain poorly defined due to the technical difficulties in specific quantification and sampling of patient liver tissue in all phases of chronic HBV infection and various states (e.g., undergoing active turnover). We believe that work in these fundamental areas of HBV research will be important to gain a broader understanding of chronic HBV infection and allow us to treat it more effectively. 

### 4.2. Implications on Therapy

Such a model has implications on the development of therapeutic strategy to cure HBV (particularly inducing the loss of circulating HBsAg, a so-called functional cure). Firstly, while it is difficult to clear integrations that are established early, preventing new infections (e.g., by reducing viral load, entry inhibition, or disrupting incoming capsids) at the appropriate points during chronic infection could limit new integrations. These therapies could also act to limit inflammation, preventing ongoing clonal expansion and amplification of existing integrations. 

If integrations indeed are causative in HCC (as many suspect), earlier interventions could limit cancer initiation by reducing new integration events and limiting clonal expansion of cell already with existing integrations. Indeed, exciting data presented at the 2020 Virtual Meeting for the American Association for the Study of Liver Diseases support these hypotheses (including Oral 16—Tenofovir Disoproxil Fumarate Treatment Reduces the Number of Transcriptionally Active Viral Integrations in Chronically Infected HBV patients, presented by Y.C. Hsu; and Abstract 22—Long term nucleos(t)ide analogue therapy reduced the extent of HBV DNA integration in chronic hepatitis B patients, presented by N. Chow). The efficacy of such strategies for HCC prevention is dependent on the relative contributions to tumorigenesis from (1) sudden clonal expansion of hepatocytes from new integrations that occur later in infections; or (2) the ongoing clonal expansion of cells with integrations established early on in infection. Therapies will be much more effective if the former is the case. 

Our model also suggests that some caution should be taken for any therapies that could induce long-lived flares (e.g., immuno-modulators). We would suggest that measurement of integration frequency be incorporated into the trial design when testing such agents. 

Finally, therapeutic modulation of HBsAg expression could alter these dynamics dramatically, given the multifaceted nature of HBsAg in this model. The aforementioned computer simulations or mathematical modelling would be ideal in identifying and designing future therapeutic strategies. 

## 5. Conclusions

In summary, the contributions of integrated HBV DNA towards HBV persistence and pathogenesis have been partially clarified and continue to be the topic of intense interest, particularly given its impact on current and upcoming therapies. Newly-developed highly controllable in vitro systems will allow the field to further dissect the roles of HBV DNA integration with greater precision in future studies.

## Figures and Tables

**Figure 1 viruses-13-00180-f001:**
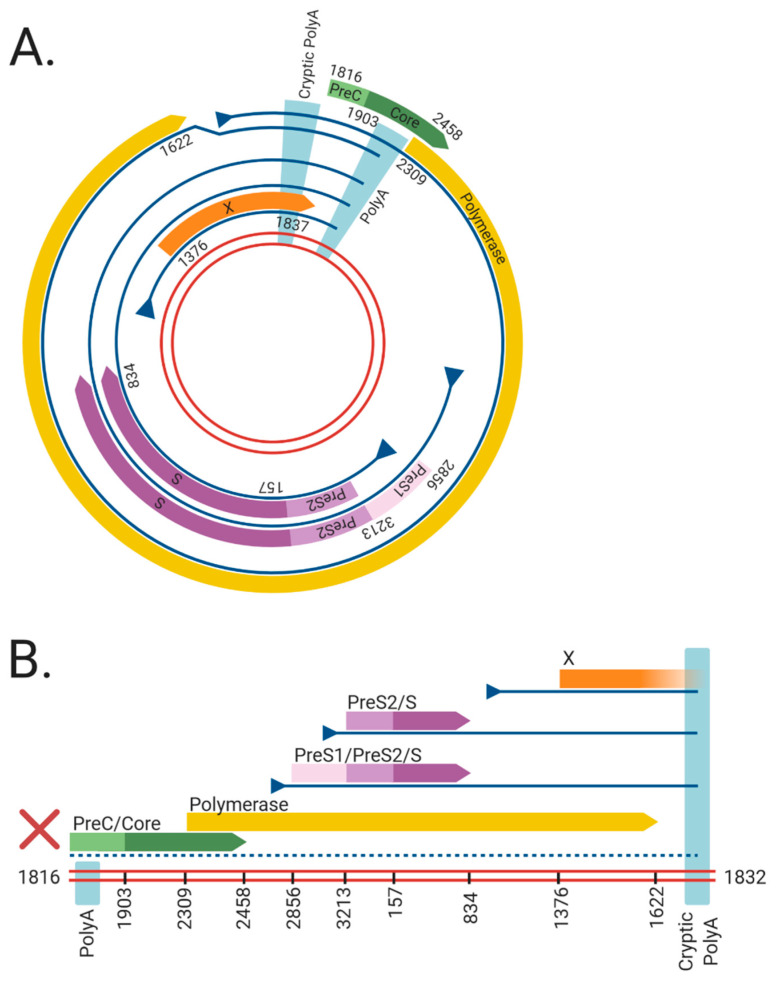
The viral RNAs and open reading frames (ORFs) expressed from the (**A**) cccDNA and (**B**) dslDNA forms of hepatitis B virus (HBV). The double red lines represent of the HBV DNA genome, blue lines represent viral mRNAs, triangles indicate transcriptional promotors, and coloured arrows represent viral ORFs. Nucleotide numbering of ORFs are shown as per Genbank Accession #AB241115. The viral mRNAs expressed from cccDNA terminate at a polyadenylation signal (poly A) located in the Core ORF. However, in dslDNA form, the canonical poly A signal is located at the 5′ end and mRNAs instead terminate at a non-canonical cryptic poly A signal. The mRNA coding for the HBV PreC/Core and Polymerase are separated from its promoter in the dslDNA form, leading to loss of their expression (dashed line). Figure generated using BioRender (https://biorender.com/).

**Figure 2 viruses-13-00180-f002:**
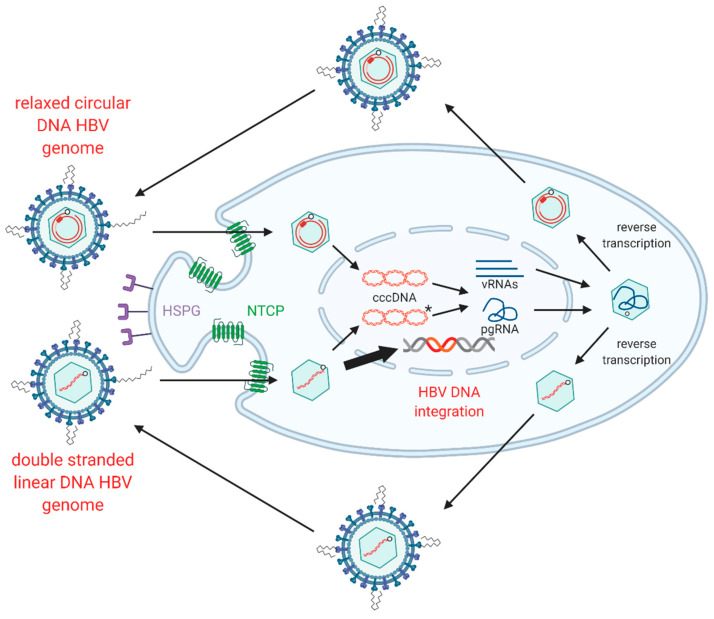
Replication cycle of hepatitis B virus (HBV) including integration into host genome. HBV virions attach to heparan sulphate proteoglycans (HSPG) on the cell surface, allowing for clathrin-mediated endocytosis via sodium taurochlorate co-transporting polypeptide (NTCP). Relaxed circular (rc)DNA genomes (top) are converted into covalently closed circular (ccc)DNA, which serve as the transcriptional template for viral mRNAs (vRNAs) and pregenomic (pg)RNA. Double stranded linear (dsl)DNA (bottom) can also form cccDNA but cannot code for functional rcDNA due to an additional 16nt insertion (asterisk) [20,21,22]. HBV pgRNA is encapsidated by viral capsid proteins and is reverse transcribed by the viral polymerase to produce either rcDNA or dslDNA. The mature nucleocapsids containing viral DNA are then enveloped by HBsAg embedded into host membranes and secreted as virions. As a secondary pathway, dslDNA can integrate into the host genome at double stranded DNA breaks, via non-homologous end joining. Integrated HBV DNA is replication-deficient due to rearrangements of the viral genome that abrogate expression of the viral polymerase and capsid proteins (Figure 1). Figure adapted from [23] and generated using BioRender (https://biorender.com/).

**Figure 3 viruses-13-00180-f003:**
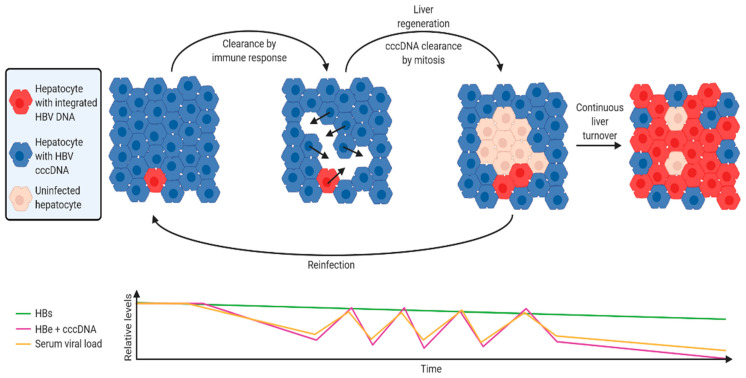
A hypothetical model of host-virus dynamics over the course of a chronic HBV infection. The majority of HBsAg is derived from cccDNA immediately after initial infection, as HBV DNA integrations are rare. When activated, the immune response preferentially targets cells expressing HBeAg and Polymerase, which are coded by cccDNA but not integrated DNA. Loss of cccDNA-containing hepatocytes also leads to clearance of cccDNA from surrounding infected hepatocytes via mitosis. Daughter cells are susceptible to new reinfection, inducing new integration events. This cycle of clearance and reinfection continues, resulting in fluctuating levels of HBeAg and cccDNA as well as serum viral load. Meanwhile, selective clonal expansion of hepatocytes with integrated HBV DNA is driven by the lack of HBeAg and polymerase expression, as well as the expression of HBsAg (rendering them refractory to new infection due to super-infection exclusion). As integrated HBV DNA frequency increases, this results in reduced HBeAg and serum viral load, while HBsAg levels remain relatively stable. Figure was generated using BioRender (https://biorender.com/).

## Data Availability

No data was generated for this publication.

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
