# Peer review of "Hepatitis B Virus DNA Integration: In Vitro Models for Investigating Viral Pathogenesis and Persistence"

_viruses, 2021, doi:10.3390/v13020180_

Round 1
Reviewer 1 Report
In this review, Thomas Tu et al., summarized in vitro models for HBV DNA integration, mechanisms of HBV integration, and HBV DNA-integration associated HCC. Tu et al. also proposed the hypothetical model of HBV DNA integration dynamic during chronic HBV infection. Overall, this review is well-written, and I have some points that need to be addressed during revision.
- It might be helpful to introduce the PreC/Core ORF in line 65. In Figure 1, authors labeled PreC/Core ORF, but in Line 65 mentioned: “four ORFs encode for the HBV core antigen (HBcAg, which forms the viral capsid)”. It could be changed to “four ORFs encode for the HBV precore/core antigen, (HBV core antigen, which forms the viral capsid; HBV precore antigen is the precursor of secretory HBV e antigen)”.
- In lines 92-93, “The mRNA coding for the HBV PreC/Core and Polymerase is separated from its promoter in the dslDNA form, leading to loss of its expression (dashed line).” Could be changed as “The mRNAs (i.e., pgRNA and preCore mRNA) coding for the HBV PreC/Core and Polymerase is separated from its promoter in the dslDNA form, leading to loss of its expression (dashed line).”
- In lines 111-112, “pre-genomic RNA is rendered non-functional as the truncation excises the epsilon RNA sequence necessary for reverse transcription.” is not a correct statement. The epsilon locates at the 5’ of dslDNA (in the PreC region, downstream of DR1), as shown in Fig.1B.
- In line 123-125, “Both relaxed circular (rc)DNA (top) and double-stranded linear (dsl)DNA (bottom) are capable of being converted into covalently closed circular (ccc)DNA, which serves as the viral template for viral mRNAs (vRNAs) and pre-genomic (pg)RNA.” is confusing. The cccDNA made from rcDNA is the templated for pgRNA. However, the cccDNA made from dslDNA is different from rcDNA, which is not competent for rcDNA synthesis due to a redundant 16 nt insertion. The authors need to clarify it to avoid confusion.
- In line 226-227, “However, HBV proteins in themselves do not appear to be carcinogenic; transgenic mice over-expressing HBV antigens are do not show any liver patho-histology [86,87].” might not be accurate. L-HBs transgenic mice show liver tumor development (Frank Chisari et al., 1989 Cell).
- In line 322-323, “If integrations indeed are causative in HCC (as many suspect), earlier interventions could limit cancer initiation through these two pathways.” Is an interesting comment. As the authors’ study using in vitro system and other groups’ studies using in vivo animals showed that DNA integration occurs early stage of infection, it might be very hard (or even impossible) to prevent HCC caused by HBV DNA integration according to the clonal expansion model. Any thoughts or comments from authors could be added.
Author Response
Reviewer 1 comments
It might be helpful to introduce the PreC/Core ORF in line 65. In Figure 1, authors labeled PreC/Core ORF, but in Line 65 mentioned: “four ORFs encode for the HBV core antigen (HBcAg, which forms the viral capsid)”. It could be changed to “four ORFs encode for the HBV precore/core antigen, (HBV core antigen, which forms the viral capsid; HBV precore antigen is the precursor of secretory HBV e antigen)”.
We have now revised these sentences for clarity:
“The HBV genome contains 4 overlapping open reading frames that encode for 7 viral proteins, including: the HBV core antigen (HBcAg, which forms the viral capsid), HBV e antigen (HBeAg, a secreted viral protein encoded by the core/pre-core ORF with reported tolerogenic properties), surface protein (HBsAg, which has 3 forms large, medium and small – or LHBs, MHBs, and SHBs), polymerase (HBV pol), and x protein (HBx, which regulates transcription of viral genes from the cccDNA episome).”
In lines 92-93, “The mRNA coding for the HBV PreC/Core and Polymerase is separated from its promoter in the dslDNA form, leading to loss of its expression (dashed line).” Could be changed as “The mRNAs (i.e., pgRNA and preCore mRNA) coding for the HBV PreC/Core and Polymerase is separated from its promoter in the dslDNA form, leading to loss of its expression (dashed line).”
We have now incorporated the change as suggested by the reviewer.
In lines 111-112, “pre-genomic RNA is rendered non-functional as the truncation excises the epsilon RNA sequence necessary for reverse transcription.” is not a correct statement. The epsilon locates at the 5’ of dslDNA (in the PreC region, downstream of DR1), as shown in Fig.1B.
We appreciate the reviewer bringing up this important point. We realise we made a mistake in this section and it is the removal of the 3’ DR1 region and phi sequences are most impactful on pgRNA reverse transcription. We have now clarified this in the revised text.
“Most viral transcripts expressed from integrated HBV DNA can retain function by terminating using a non-canonical poly A signal retained at its 3’ end [30, 31]. Howev-er, pre-genomic RNA is rendered non-functional as the premature termination re-moves structural elements from the 3’ end necessary for initiating pgRNA reverse transcription (namely, the direct repeat 1 and phi sequences [32-35]).”
In line 123-125, “Both relaxed circular (rc)DNA (top) and double-stranded linear (dsl)DNA (bottom) are capable of being converted into covalently closed circular (ccc)DNA, which serves as the viral template for viral mRNAs (vRNAs) and pre-genomic (pg)RNA.” is confusing. The cccDNA made from rcDNA is the templated for pgRNA. However, the cccDNA made from dslDNA is different from rcDNA, which is not competent for rcDNA synthesis due to a redundant 16 nt insertion. The authors need to clarify it to avoid confusion.
We have now added extra details in this section to clarify this phenomenon:
“Relaxed circular (rc)DNA genomes (top) are converted into covalently closed circular (ccc)DNA, which serve as the transcriptional template for viral mRNAs (vRNAs) and pre-genomic (pg)RNA. Double stranded linear (dsl)DNA (bottom) can also form cccDNA, but cannot code for functional rcDNA due to an additional 16nt insertion [43-45].”
In line 226-227, “However, HBV proteins in themselves do not appear to be carcinogenic; transgenic mice over-expressing HBV antigens are do not show any liver patho-histology [86,87].” might not be accurate. L-HBs transgenic mice show liver tumor development (Frank Chisari et al., 1989 Cell).
We thank the reviewer for catching our mistake. We were incorrect in saying over-expression in this context: the cited models express HBV under native promoters. We have now corrected this and added in the citation mentioned for clarification.
“However, HBV proteins in themselves do not appear to be carcinogenic; transgenic mice expressing HBV antigens from native promoters do not show any liver patho-histology [93, 94].”
“Similarly, over-expression of HBsAg from a highly transcriptionally-active albumin promoter spontaneously induces pre-neoplastic lesions and HCC in transgenic mice [99], while mice expressing HBsAg alone from native promoters maintain histologically normal livers [100, 101].”
In line 322-323, “If integrations indeed are causative in HCC (as many suspect), earlier interventions could limit cancer initiation through these two pathways.” Is an interesting comment. As the authors’ study using in vitro system and other groups’ studies using in vivo animals showed that DNA integration occurs early stage of infection, it might be very hard (or even impossible) to prevent HCC caused by HBV DNA integration according to the clonal expansion model. Any thoughts or comments from authors could be added.
We appreciate the reviewer bringing up this point. We have clarified our proposal in the revised version of this section:
“If integrations indeed are causative in HCC (as many suspect), earlier interventions could limit cancer initiation by reducing new integration events and limiting clonal expansion of cell already with existing integrations. Indeed, exciting preliminary data presented at the 2020 Virtual Meeting for the American Association for the Study of Liver Diseases support these hypotheses (including Oral 16 - Tenofovir Disoproxil Fumarate Treatment Reduces the Number of Transcriptionally Active Viral Integrations in Chronically Infected HBV patients, presented by Y.C. Hsu; and Abstract 22 - Long term nucleos(t)ide analogue therapy reduced the extent of HBV DNA integration in chronic hepatitis B patients, presented by N. Chow). The efficacy of such strategies for HCC prevention is dependent on the relative contributions to tumorigenesis from: 1) sudden clonal expansion of hepatocytes from new integrations that occur later in infections; or 2) the ongoing clonal expansion of cells with integrations established early on in infection. Therapies will be much more effective if the former is the case.”
Reviewer 2 Report
Dear authors,
the manuscript entitled "Hepatitis B virus DNA integration: in vitro models for investigating viral pathogenesis and persistence" is, in general, well-written and may represent for the Journal readership.
I have some concerns regarding the manuscript.
- Please, remove some references on your own works, or enlarge the number of studies considered in the manuscript.
- Figure 2 can be easily removed since it contains too common information. Otherwise, please, specify in the text how this Figure helps understanding the process of HBV dslDNA and HBV DNA integration.
- Please, add the description of the web-databases, where the information about various in vitro experiments for HBV DNA integration can be found. Please, add more detailed information about each method and its reproducibility in general. Please, specify the Tumour-derived cell line, which are used in in vitro system. Are they the same for PLC/PRF/5 ,Hep3B T, HepG2.2.15, HepAD38? Meanwhile, the footnote for (1) is absent (Table 1).
- Please, add the take-home message of the review. What are the adcantages and disadvantages of each in vitro method. Why A hypothetical model of HBV DNA integration dynamics during chronic infection should have an impact on the HBV treatment and cure. Are there any other models of HBV DNA integration dynamics during chronic infection in literature. Which experiments/prospective studies should be performed to support the model suggested.
Kind regards.
Author Response
Reviewer 2
1.Please, remove some references on your own works, or enlarge the number of studies considered in the manuscript.
As our group is one of the few investigating HBV integration in vitro, it would limit the review dramatically if we were to remove these studies. To address this issue, we have extensively rewritten section 2.2., expanding the scope of works by discussing in detail the contribution of in vitro systems of other closely-related hepadnaviruses (duck HBV and woodchuck hepatitis virus). We have now expanded the number of references by 12. We also realise 4 of our group’s references were actually repeated in the bibliography due to a mismatch in Endnote libraries; we have now corrected this. We hope that these measures are sufficient to address the reviewer’s concerns and we are open to consider any additional references that the reviewer thinks should be added to the review.
2.Figure 2 can be easily removed since it contains too common information. Otherwise, please, specify in the text how this Figure helps understanding the process of HBV dslDNA and HBV DNA integration.
The scope of this special issue “Virus Integration” includes all viruses that integrate, and not just HBV. We believe that a brief overview of the HBV replication cycle (and how HBV DNA integration occurs with that context) is necessary to provide the non-specialist reader some understanding of the bulk of the review.
3.Please, add the description of the web-databases, where the information about various in vitro experiments for HBV DNA integration can be found. Please, add more detailed information about each method and its reproducibility in general. Please, specify the Tumour-derived cell line, which are used in in vitro system. Are they the same for PLC/PRF/5 ,Hep3B T, HepG2.2.15, HepAD38? Meanwhile, the footnote for (1) is absent (Table 1).
4.Please, add the take-home message of the review. What are the adcantages and disadvantages of each in vitro method. Why A hypothetical model of HBV DNA integration dynamics during chronic infection should have an impact on the HBV treatment and cure. Are there any other models of HBV DNA integration dynamics during chronic infection in literature. Which experiments/prospective studies should be performed to support the model suggested.
Unfortunately, the majority of in vitro experiment sequence data are not deposited in centralised locations or online databases. We have provided the references to the papers, which direct users to this information where available (e.g. as supplementary data or specific Genbank accession numbers).
Methods for detecting HBV DNA integrations is a rather technical aspect that we believe takes away from the focus of the paper. We have directed readers to a more technical review that has covered this subject prior (Line 174).
We have now added a section providing more information on the in vitro models and discussing their relative advantages/disadvantages:
“In vitro models vary in their ease of use (or reproducibility) in integration studies and inversely how closely they resemble integrations from a true infection in people with HBV. Tumour-derived cell lines from HBV patients contain integrations formed in the native setting, but cannot be used to understand underlying mechanisms as they do not generate new integrations. Conversely, nuclear introduction of large numbers of HBV DNA molecules (via over-expressing constructs or transfection) can drive relatively high integration rates, allowing easy study of the integration process. However, this does not reflect a true infection; nuclear import of new HBV virions occurs relatively inefficiently in HBV infection [47-50], so levels of HBV DNA in infected hepatocytes is relatively low. Newly-developed infection models expressing the HBV receptor have solved this to some degree by largely recapitulating the virological aspects accurately. However, the low integration rate is challenging to detect without specialised methods, which can be difficult to reproduce. In essence, all studies need to be inter-preted with these limitations in mind.”
Regarding the specifying tumour-derived cell line and the missing footnote for Table 1, post-submission formatting by the journal staff had rearranged the table so that it relocated the footnote for 1 above the table. We have since relocated it and hope this clarifies the issue raised by the reviewer.
We have also expanded the section on clinical implications of the hypothetical model (as outlined in our response to Reviewer 1).
To our knowledge, there are no other overarching models like this incorporating HBV DNA integration, clonal expansion and other features we have discussed into the framework of a chronic HBV infection. We have added an additional section to make this point:
“While many models have been developed to try to understand viral-host dynamics [121], none of these appear to have incorporated the effects of HBV DNA integration, clonal expansion and other new phenomena that have recently been discovered in recent studies.”
We have now added a section outlining potential future experimental directions to pursue in the hopes of confirming our hypothetical model:
“There are several aspects to this theoretical model that require further experimental support. Mathematical modelling would be ideal for determining if emergent behaviour from these initial assumptions and multiple interacting factors reflects clinical observations of HBV patients. To provide realistic ranges for parameters, precise experiments must be carried out to define the effect sizes of each of these factors (e.g. immunosuppression by HBsAg, immune-stimulation by HBeAg-expressing cells). This may require a firmer understanding of anti-HBV immunity than we have currently.
Even relatively simple parameters, such as cccDNA per hepatocyte, remain poorly defined due to the technical difficulties of specific quantification and sampling of patient liver tissue in all phases of chronic HBV infection and various states (e.g. undergoing active turnover). We believe that work in these fundamental areas of HBV research will be important to gain a broader understanding of chronic HBV infection and allow us to treat it more effectively.”
Round 2
Reviewer 2 Report
Dear authors,
I can see that some my comments were taken into account, the quality of the manuscript is increased in my point of view.
I still recommend double-cheking English, Figures and References before publication.
Best regards.